# COUNTERFACTUAL SELF-TRAINING

## ABSTRACT

Unlike traditional supervised learning, in many settings only partial feedback is available. We may only observe outcomes for the chosen actions, but not the counterfactual outcomes associated with other alternatives. Such settings encompass a wide variety of applications including pricing, online marketing and precision medicine. A key challenge is that observational data are influenced by historical policies deployed in the system, yielding a biased data distribution. We approach this task as a domain adaptation problem and propose a self-training algorithm which imputes outcomes with *finite discrete* values for *finite* unseen actions in the observational data to simulate a randomized trial. We offer a theoretical motivation for this approach by providing an upper bound on the generalization error defined on a randomized trial under the self-training objective. We empirically demonstrate the effectiveness of the proposed algorithms on both synthetic and real datasets.

## 1 INTRODUCTION

Counterfactual inference (Pearl et al., 2000) attempts to address a question central to many applications - *What would be the outcome had an alternative action was chosen?* It may be selecting relevant ads to engage with users in online marketing (Li et al., 2010), determining prices that maximize profit in revenue management (Bertsimas & Kallus, 2016), or designing the most effective personalized treatment for a patient in precision medicine (Xu et al., 2016). With observational data, we have access to past actions, their outcomes, and possibly some context, but in many cases not the complete knowledge of the historical policy which gave rise to the action (Shalit et al., 2017). Consider a pricing setting in the form targeted promotion. We might record information of a customer (context), promotion offered (action) and whether an item was purchased (outcome), but we do not know why a particular promotion was selected.

Unlike traditional supervised learning, we only observe feedback for the chosen action in observational data, but not the outcomes associated with other alternatives (i.e., in the pricing example, we do not observe what would occur if a different promotion was offered). In contrast to the gold standard of a randomized controlled trial, observational data are influenced by historical policy deployed in the system which may over or under represent certain actions, yielding a biased data distribution. A naive but widely used approach is to learn a machine learning algorithm directly from observational data and use it for prediction. This is often referred to as direct method (DM) (Dudík et al., 2014). Failure to account for the bias introduced by historical policy often results in an algorithm which has high accuracy on the data it was trained on, but performs considerably worse under a different policy. For example in the pricing setting, if historically most customers who received high promotion offers bear a certain profile, then a model based on direct method may fail to produce reliable predictions on these customers when low offers are given.

To overcome the limitations of direct method, Shalit et al. (2017); Johansson et al. (2016); Lopez et al. (2020) cast counterfactual learning as a *domain adaptation* problem, where the source domain is observational data and the target domain is a *randomized trial* whose assignment of actions follows a uniform distribution for a given context. The key idea is to map contextual features to an embedding space and jointly learn a representation that encourages similarity between these two domains, leading to better counterfactual inference. The embedding is generally learned by a neural network and the estimation of the domain gap is usually slow to compute.

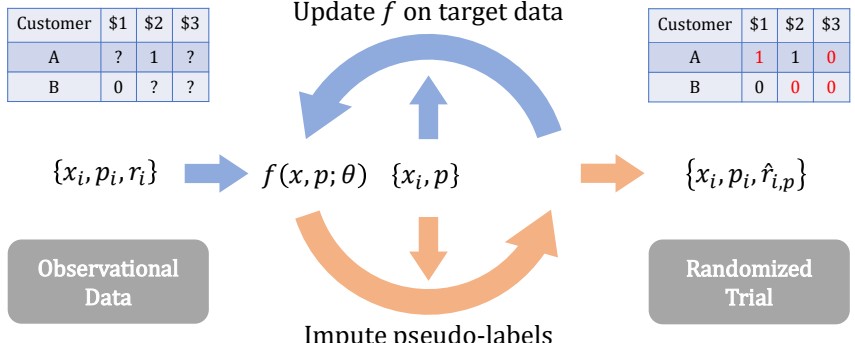

Figure 1: Illustration of the proposed Counterfactual Self-Training (CST) framework. There are two sales records (observational data) shown in the table, *i.e.*, , Customer A was offered $2 and bought an item; Customer B was offered $1 and did not buy. The question marks in the tables represent the counterfactual outcome which we do not observe. For all these unseen counterfactual outcomes, pseudo-labels which are colored in red in the tables are imputed by a model and are used to augment the observational data. The model is subsequently updated by training on both the imputed counterfactual data and the factual data. This iterative training procedure continues until it converges.

In this paper, while we also view counterfactual inference as a domain adaptation problem between observational data and an ideal randomized trial, we take a different approach - instead of estimating the domain gap between the two distributions via an embedding, we explicitly simulate a randomized trial by imputing pseudo-labels for the unobserved actions in the observational data. The optimization process is done by iteratively updating the pseudo-labels and a model that is trained on both the factual and the counterfactual data, as illustrated in Figure 1. As this method works in a self-supervised fashion (Zou et al., 2018; Amini & Gallinari, 2002), we refer to our proposed framework as Counterfactual Self-Training (CST).

The contribution of our paper is as follows. First, we propose a novel self-training algorithm for counterfactual inference. To the best of our knowledge, this is the first application of self-training algorithm for learning from observational data. Moreover, in contrast to the existing methods from domain adaption on counterfactual inference, CST is flexible and can work with a wide range of machine learning algorithms, not limited to neural networks. Second, we offer a theoretical motivation of our approach by providing an upper bound on the generalization error defined on a randomized trial under the self-training objective. In other words, we show that the counterfactual self-training algorithm helps minimizing the risk on the target domain. Our theoretical bounds suggest generating pseudo-labels with random imputation, which is a methodological departure from traditional self-training algorithms which impute hard labels. Third, we present comprehensive experiments on several synthetic datasets and three counterfactual learning datasets converted from multi-label classification tasks to evaluate our method against state-of-the-art baselines. In all experiments, CST shows competitive or superior performance against all the baselines. Moreover, our algorithm is easy to optimize with a much faster training time than other baselines.

## 2 RELATED WORK

Counterfactual policy optimization has received a lot of attention in the machine learning community in the recent years (Swaminathan & Joachims, 2015a; Joachims et al., 2018; Shalit et al., 2017; Lopez et al., 2020; Kallus, 2019; Kallus & Zhou, 2018; Wang et al., 2019). Most of the proposed algorithms can be divided into two categories: counterfactual risk minimization (CRM) and direct method (DM). Both can be used together to construct doubly robust estimators (Dudík et al., 2014) to further improve efficiency. CRM, also known as off-policy learning or batch learning from bandit feedback, typically utilizes inverse propensity weighting (IPW) (Rosenbaum, 1987; Rosenbaum & Rubin, 1983) to account for the bias in the data. Swaminathan & Joachims (2015a) introduces the CRM principle with a variance regularization term derived from an empirical Bernstein bound (Maurer & Pontil, 2009) for finite samples. In order to reduce the variance of the IPW

estimator, Swaminathan & Joachims (2015b) proposes a self-normalized estimator, while Bandit-Net (Joachims et al., 2018) utilizes the baseline technique (Greensmith et al., 2004) in deep nets. As pointed out by Lefortier et al. (2016), CRM-based methods tend to struggle with medium to large action spaces in practice. Morever, CRM-based methods generally require a known and stochastic logging policy, along with full support on the action space. When either one of the requirements is violated, Sachdeva et al. (2020); Kang et al. (2007) observe direct method often demonstrates a more robust performance. When the logging policy is not available, the counterfactual learning problem is often referred to as learning from observational data, which is the setting we focus on. In addition to select the optimal actions, direct method can also be used to identify causal treatment effect (Künzel et al., 2019), CST can be viewed as an extention to direct method.

Learning from observational data is also closely related to estimating Individualized Treatment Effects (ITE) (Shpitser & Pearl, 2012) or conditional average treatment effect (CATE), which is defined as the difference of expected outcomes between two actions, with respect to a given context. The main challenge of identifying ITE is that unlike an ideal randomized trial, observational data is biased and we do not have the access to the counterfactuals. Hill (2011) uses a bayesian nonparametric algorithm to address this issue. Yoon et al. (2018) proposes using generative adversarial nets to capture the uncertainty in the counterfactual distributions to facilitate ITE estimation. Johansson et al. (2016); Shalit et al. (2017) approach counterfactual inference with representation learning and domain adaptation. Their key idea is to learn a representation between observational data and a randomized trial that encourages better generalization on all possible actions. It is achieved by minimizing a weighted sum of the factual loss on the observational data (loss for direct method) plus an estimated domain gap measured by integral probability metrics. Lopez et al. (2020) further extends this framework to multiple treatments using Hilbert-Schmidt Independence Criterion (HSIC) (Gretton et al., 2008) and achieves state-of-the-art performance. The HSIC proposed in Lopez et al. (2020) has a computation time of at least $\mathcal{O}(N^2)$, making its training process slow. While the aforementioned methods and our approach can be viewed as extensions to direct method, we tackle the domain adaptation problem differently by explicitly augmenting the observational data to create a simulated randomized trial via self-training. Different counterfactual estimation algorithms are classified as X-, T-, S-learner in Künzel et al. (2019), for example Hill (2011) is an instance of S-learner. Our approach is similar to X-learner which uses pseudo-label to create counterfactuals, but CST considers multiple instead of binary actions and is trained in an iterative fashion.

Self-training algorithms have been widely studied in semi-supervised learning and domain adaptation problems (Nigam et al., 2000; Amini & Gallinari, 2002; Grandvalet & Bengio, 2005; Zou et al., 2019; Han et al., 2019). Grandvalet & Bengio (2005) proposes to use entropy regularization for semi-supervised learning as a class-overlapping measure to utilize unlabeled data. Nigam et al. (2000); Amini & Gallinari (2002); Zou et al. (2019) formulate the pseudo-label imputation as classification EM algorithm and show its convergence under proper initialization. Han et al. (2019) points out that pseudo-label imputation can be viewed as minimizing min-entropy as a type of Rényi entropy $\frac{1}{1-\alpha} \log(\sum_{i=1}^{n} p_i^{\alpha})$ when $\alpha \to \infty$, and Shannon entropy in Grandvalet & Bengio (2005) is the case when $\alpha \to 1$. Self-training is also shown to be effective in semi-supervised learning for many other machine learning models besides neural networks (Tanha et al., 2017; Li et al., 2008). It is worthy to mention that unlike traditional self-training where the target domain is given by the problem, we propose to construct a target domain by imputing pseudo-labels on all unseen actions to simulate a pseudo-randomized trial. Moreover, instead of hard labels used in traditional self-training, we propose to use random imputation to create pseudo-labels which have a theoretical motivation tailored for counterfactual inference and are shown to be more effective based on the experiments results.

## 3 PROBLEM STATEMENT

Following the notation in Lopez et al. (2020), we use $\mathcal{X}$ to represent an abstract space and $\mathbb{P}(x)$ is a probability distribution on $\mathcal{X}$. Each sample $x = x_1, \cdots, x_n \in \mathcal{X}^n$ is drawn independently from $\mathbb{P}(x)$. $\mathcal{P}$ is the *discrete* action space that a central agent can select for each sample, after which a *discrete* reward $r$ with finite possible values is revealed to the agent. In precision medicine, $\mathcal{X}$ may represent a patient cohort, $\mathcal{P}$ refers to feasible treatment for a disease, and $r$ can be the indicator of whether a patient survives after the treatment. Similarly, $\mathcal{X}, \mathcal{P}, r$ can represent visitors, ads shown and whether visitor clicks in online marketing.

We focus on an example of pricing to illustrate our method. We use $x \in \mathcal{X}^n \sim \mathbb{P}(x)$ to denote a customer. Let $\mathcal{P}$ represent finite price options a central agent can offer to customers. After offering price $p \in \mathcal{P}$, the agent observes the response from the customer $r \sim \mathbb{P}(r|x, p)$, *i.e.,* , either a 1 (buy) or a 0 (no-buy). As a direct method, the task is to learn a function $f(x, p)$ by minimizing the loss $\mathbb{E}_{x \sim \mathbb{P}(x), p \sim \pi_0(p|x)} \mathcal{L}(f(x, p), r)$ where $\pi_0(p|x)$ is a randomized assignment policy (Shalit et al., 2017; Lopez et al., 2020). The estimation task is often referred to as demand estimation (Wales & Woodland, 1983), which is critical for many downstream decisions such as inventory optimization and revenue management (Kök & Fisher, 2007; McGill & Van Ryzin, 1999). This is in contrast to CRM-based methods which use the final reward as objective to learn a policy $\pi(p|x)$ that maximizes $\mathbb{E}_{x \sim \mathbb{P}(x), p \sim \pi(p|x)} \mathbb{E}[r|x, p]$ (Swaminathan & Joachims, 2015a).

With observational data, the individualized treatment effect is not always identifiable. We use Rubin's potential outcome framework and assume consistency and strong ignorability which is a sufficient condition for identifying ITE (Imbens & Wooldridge, 2009; Pearl, 2017). Here we formally present the ignorability assumption (Rubin, 2005; Shalit et al., 2017):

**Assumption 3.1** (Ignorability). *Let $\mathcal{P}$ be action set, $x$ is context (feature), $r(p)|x$ is observed reward for action $p \in \mathcal{P}$ given context $x$, $r(p) \perp\!\!\!\perp p|x, \forall p \in \mathcal{P}$.*

In other words, we assume there is no unobserved confounders. This condition generally cannot be made purely based on data and requires some domain knowledge.

## 4 ALGORITHM

In this section, we introduce Counterfactual Self-Training (CST) algorithm, which can be viewed as an extension of the direct method via domain adaptation. Unlike existing methods using representation learning, we propose a novel self-training style algorithm to account the bias inherent in the observational data.

### 4.1 SELF-TRAINING

Self-training has recently been used in unsupervised domain adaptation (UDA) and semi-supervised learning (SSL) and achieved great success (Zou et al., 2019; Han et al., 2019; Zou et al., 2018; Amini & Gallinari, 2002; Nigam et al., 2000; Grandvalet & Bengio, 2005). The self-training algorithm works in an iterative fashion: First, after training a classifier $f(x, p)$ on a source dataset, pseudo-labels are created by the best guess of $f$. Next, the model is trained on a target dataset, and the trained model is used to generate new pseudo labels. This idea is illustrated in Figure 1.

To formulate the counterfactual learning problem as a domain adaptation problem, observational data is viewed as data sampled from a source distribution $\mathcal{D}_S = \mathbb{P}(x)\pi(p|x)$. The target domain is a randomized trial on the same feature distribution to ensure a uniformly good approximation on all actions. Our goal is to transfer observational data from the source domain to a simulated pseudo-randomized trial via self-training. To accomplish this, we first train an initial classifier $f_0(x, p)$ on observational data, then impute pseudo-labels on all unseen actions from the observation data with $\hat{r}_{i,p} \sim f(x_i, p)$. The model is then updated by training with the following objective:

$$\min_{\theta} \mathcal{L}_{ST} = \frac{1}{N|\mathcal{P}|} \Big( \underbrace{\sum_{i=1}^{N} l(f_\theta(x_i, p_i), r_i)}_{\mathcal{L}_{src}} + \sum_{i=1}^{N} \sum_{p \in \mathcal{P} \setminus p_i} l(f_\theta(x_i, p), \hat{r}_{i,p}) \Big) \tag{1}$$

The first term $\mathcal{L}_{src}$ in Equation 1 corresponds to the loss used in direct method, defined over the factual data alone. Meanwhile, the second term refers to the loss defined over the imputed counterfactual data. In other words, in order to get a good model across all actions, we utilize the pseudo-population induced from imputation which represents a simulated randomized trial. We iteratively train the model and impute pseudo-labels until it converges. The algorithm is stated in Algorithm 1.

Note that a key difference between our CST algorithm and traditional self-training (ST) methods for unsupervised domain adaptation (such in Zou et al. (2018)): Pseudo-labels in traditional ST are

---

**Algorithm 1** Counterfactual Self-Training

---

1: **while** NOT converged **do** ▷ Main training loop
2:     **for** each $i \in \{1 \dots N\}$ **do**
3:         **for** each $p \in \mathcal{P} \setminus p_i$ **do**
4:             Impute pseudo-label $\hat{r}_{i,p} \sim f_\theta(r|x_i, p)$. ▷ Pseudo-label imputation
5:         **end for**
6:     **end for**
7:     Update $\theta$ by minimizing $\mathcal{L}_{ST}$ defined in Equation 1. ▷ Self-training
8: **end while**

---

generated from hard imputation while ours are sampled from a probability distribution as illustrated in Algorithm 1 line 4. Not only this randomized imputation has a theoretical motivation presented in Section 4.2, it also demonstrates superior performance over hard imputation in our experiments in Section 5.

## 4.2 THEORETICAL MOTIVATION

As our objective is to augment observational data to a randomized trial such that the learnt model is able to perform better on all feasible actions, we focus on bounding the generalization error defined on a randomized trial. We use $\mathcal{D}$ to represent the distribution of a true randomized trial where the assignment policy is a uniform probability over $\mathcal{P}$ given context, and $\hat{\mathcal{D}}$ is the distribution of pseudo-label generated by the current model output $f_\theta(r|x, p)$. Define $\mathcal{R}_{\mathcal{D}}(f)$ as the risk of function $f$ with respect to a loss function $l(\cdot, \cdot)$ as $\mathcal{R}_{\mathcal{D}}(f) = \mathbb{E}_{x,p \sim \mathcal{D}}[l(f(x, p), p)]$, and $\hat{\mathcal{R}}_{\hat{\mathcal{D}}}(f)$ as empirical risk on $\hat{\mathcal{D}}$. Assume our classifier outputs a probability estimation $f_\theta(r|x, p)$ for a feature and action pair $(x, p)$, and we use a random imputation $\hat{r} \sim f_\theta(r|x, p)$ to generate outcomes for the unseen actions. We have the following theorem on the generalization bound:

**Theorem 1.** *Assume $f_\theta(r|x, p) \geq \frac{1}{M_0+1}$, where $M_0 > 1$ is constant, $x, p$ is defined on a compact, discrete set respectively, let $M = \min\{\max_{x,p}(\frac{\mathbb{P}}{f_\theta} - 1), M_0\}$, $f^\star = \operatorname{argmin}_{f \in \mathcal{F}} \mathcal{R}_{\mathcal{D}}(f)$, $\hat{\mathcal{D}}$ is the dataset generated by random imputation of current model output $f_\theta$, and $\hat{f}$ minimizes the empirical risk on $\hat{\mathcal{D}}$. For any loss function $l(\cdot, \cdot)$, we have:*

$$\mathcal{R}_{\mathcal{D}}(\hat{f}) - \mathcal{R}_{\mathcal{D}}(f^\star) \leq C(\sqrt{\frac{V}{n}} + \sqrt{\frac{\log(1/\delta)}{n}}) + (M+1)\hat{\mathcal{R}}_{\hat{\mathcal{D}}}(\hat{f}) - \mathcal{R}_{\mathcal{D}}(f^\star) \tag{2}$$

$V$ is the VC dimension of hypothesis class $\mathcal{F}$, and $C$ is a universal constant. The proof is in Appendix A.1. By replacing $M$ with $M_0$ and minimizing the right hand side of Equation 2 over $\theta$, we recover Equation 1 which is the objective that we are optimizing in the training procedure. This complete optimization involves optimizing over $\theta$ and $\hat{r}$, and can be solved via classification EM (CEM) (Amini & Gallinari, 2002) and traditional self-training is an instance of CEM (Zou et al., 2019). These methods use a hard label as a classification step to impute the pseudo-labels but it is not clear how it relates to the risk that we are interested in. To establish Theorem 1, we require a random imputation of labels based on the probability output of the classifier to upper bound the risk under a true randomized trial using this objective. Therefore, we use a random sampling to generate pseudo-labels in our algorithm, and it is shown to be more robust than hard labels in our experiments. We would like to note this bound is relatively loose when $\mathbb{P}$ is very different from $f_\theta$, thus we only use it as a motivation of our proposed algorithm. Since in the source domain, the data is generated by the true data distribution, it is possible to get a tighter upper bound, which we leave for future work.

Our assumption $f_\theta(r|x, p) \geq \frac{1}{M_0+1}$ is also proposed in Zou et al. (2019) in the form of entropy regularization to prevent the model converging too fast and getting too confident in the early stages of training. Since cross-validation is biased in the counterfactual learning due to the logging policy (Lopez et al., 2020), we avoid using hyperparameters and do not use this regularization in our experiments. Note that in traditional semi-supervised learning, deterministic pseudo-labels are commonly used by *argmax operation* $\hat{r} = \operatorname{argmax}_r f(r|x, p)$, which we refer to as CST-AI, and we refer the one with *random imputation* as CST-RI.

In Proposition 1, we show the convergence result of CST-AI . Theoretical analysis on convergence of CST-RI is more challenging. However, we empirically observe CST-RI converges without any additional techniques in all of our experiments. We show empirical loss curves and discuss the connection between CST-RI and entropy regularization (Grandvalet & Bengio, 2005) in Section A.3 in Appendix.

**Proposition 1.** *CST with argmax imputation is convergent under certain conditions.*

*Proof.* Please refer to Section A.2 of Appendix. □

## 5 EXPERIMENTS

We construct synthetic datasets for a pricing example and utilize three real datasets to demonstrate the efficacy of our proposed algorithm. Implementationwise, we use a three layer neural network with 128 nodes as our model and binary entropy loss as the loss function. We avoid using early stopping and train each method until convergence to ensure a fair comparison. The following baselines are considered in our experiments:

- Direct Method (DM): This baseline directly trains a model on observational data.
- HSIC (Lopez et al., 2020): We use the last layer as an embedding and calculate HSIC between the embedding and the actions. The training objective is binary cross entropy loss + $\lambda$·HSIC, where $\lambda$ is the hyperparameter which we choose from a grid search over $[0.01, 0.1, 1, 10, 100]$.
- BanditNet (Joachims et al., 2018): BanditNet is a CRM-based method developed for deep nets. For the baseline required in BanditNet, we normalize the reward as in Swaminathan & Joachims (2015a) and choose the hyperparameter using a grid search over $[0, 0.2, 0.4, 0.6, 0.8]$ and cross validation. We fit an additional logging policy model on historical data for BanditNet.
- Uniform DM (Wang et al., 2019): Uniform DM (UDM) also estimates the logging policy using historical data and use importance sampling to simulate a randomize trial.

Since BanditNet is designed for reward maximization, evaluation of the accuracy (*i.e.*, hamming loss) is not appropriate under our problem. In each experiment, we only evaluate BanditNet in the reward comparison. We also experiment with two versions of CST, CST-AI and CST-RI. Unlike CST, HSIC and BanditNet require a hyperparameter as an input to their algorithms. Following Joachims et al. (2018); Lopez et al. (2020), we use a 5-fold cross-validation and grid search to select the hyperparameter for all experiments. All experiments are conducted using one NVidia GTX 1080-Ti GPU with five repetitions. Mean and standard error are reported for each metric.

### 5.1 SYNTHETIC DATASETS

In synthetic experiments, we use a pricing example similar to the experiment in Lopez et al. (2020). Let $U(\cdot, \cdot)$ be a uniform distribution. Assume customer features are a 50-dimensional vector $X$ drawn from $U(0, 1)^{50}$ and there are 10 price options from \$1 to \$10. The logging policy is set as $\pi(p = i|x) = \frac{x_i}{\sum_{i=1}^{10} x_i}$. Five types of demand functions are simulated, and the complete data generation process is detailed in Appendix A.5.

We generate 1000 samples for each demand function and report hamming loss which relies on the hard labels generated by the algorithm in Table 1. In addition, as we are interested in probability estimation, we report the multi-label soft margin loss in Table 2. Lastly, as a pricing application, we also evaluate the revenue generated on the test set by solving the revenue maximization problem:

$$p_i = \underset{p}{\operatorname{argmax}} \ \mathbb{P}(r = 1|x_i, p) \cdot p \tag{3}$$

For each dataset, the test set has 5000 samples from the corresponding demand distribution. The results are shown in Table 3.

Among all datasets, CST-RI has the best performance in terms of both hamming loss and soft margin loss. HSIC outperforms DM baseline by a significant margin and comes as a close second to CST-RI. In 4 out of 5 demand functions (with the exception of D1), CST-RI achieves a comparable

or superior performance on reward as shown in Table 3. Hence, while CST-RI results in the best demand model in terms of the losses, it does not guarantee the highest revenue in all cases. This is because the downstream optimization task is independent from demand estimation (Elmachtoub & Grigas, 2017). Nevertheless, CST-RI significantly outperforms BanditNet which is designed for reward maximization due to unknown logging policy (Kang et al., 2007). We also want to point out that CST-AI performs worse than DM which is a naive baseline, demonstrating the importance of random imputation in our algorithm.

| | D1 | D2 | D3 | D4 | D5 |
|---|---|---|---|---|---|
| Direct Method | 0.2101±0.0042 | 0.2418±0.0026 | 0.3358±0.0036 | 0.2614±0.0051 | 0.1753±0.0024 |
| CST-RI | **0.1766**±0.0020 | **0.1833**±0.0030 | **0.2716**±0.0049 | 0.2000±0.0026 | **0.1447**±0.0029 |
| CST-AI | 0.4342±0.0699 | 0.5568±0.0498 | 0.4544±0.0432 | 0.5514±0.0507 | 0.4769±0.0533 |
| HSIC | 0.1940±0.0036 | 0.1947±0.0023 | 0.2821±0.0078 | **0.1969**±0.0036 | 0.1447±0.0029 |
| UDM | 0.2246±0.0053 | 0.2687±0.0114 | 0.2951±0.0059 | 0.2059±0.0049 | 0.1892±0.0145 |

Table 1: Hamming Loss on Synthetic Dataset.

| | D1 | D2 | D3 | D4 | D5 |
|---|---|---|---|---|---|
| Direct Method | 1.8032±0.1118 | 2.2517±0.1760 | 3.4890±0.2734 | 3.2841±0.1644 | 1.9047±0.1838 |
| CST-RI | **0.3729**±0.0047 | **0.4134**±0.0020 | **0.5230**±0.0044 | **0.4319**±0.0019 | **0.3206**±0.0026 |
| CST-AI | 1.2618±0.1684 | 1.5679±0.1181 | 1.3896±0.0988 | 1.5534±0.1216 | 1.3196±0.1259 |
| HSIC | 0.4407±0.0134 | 0.4393±0.0088 | 0.5578±0.0077 | 0.4405±0.0064 | 0.3284±0.0050 |
| UDM | 0.9086±0.0571 | 1.0824±0.0171 | 1.0082±0.0317 | 0.8481±0.0293 | 0.8152±0.0482 |

Table 2: Multi-Label Soft-Margin Loss on Synthetic Datasets

| | D1 | D2 | D3 | D4 | D5 |
|---|---|---|---|---|---|
| Direct Method | **4861.6**±149.5 | 4240.2±108.9 | **7806.6**±141.5 | 4442.6±124.3 | 3238.6±113.8 |
| CST-RI | 3894.4±118.2 | **4626.8**±265.1 | 7622.0±208.0 | **4692.6**±295.3 | **3381.6**±239.8 |
| CST-AI | 3800.6±113.6 | 2212.4±138.4 | 6975.2.4±217.6 | 2084.4±128.2 | 1082.6±40.7 |
| HSIC | 3692.0±76.3 | 4361.4±280.8 | 7416.0±276.4 | 4653.8±219.4 | 3050.4±76.2 |
| BanditNet | 3610.4±106.4 | 4334.6±144.7 | 6739.2±112.7 | 4582.0±335.9 | 2907.2±222.2 |
| UDM | 3580.8±94.9 | 4430.0±264.4 | 6014.6±235.2 | 4208.2±482.5 | 1844.8±416.8 |
| Oracle | 6505.2±158.8 | 5082.0±0 | 9527.6±231.0 | 5904.4±129.2 | 4423.4±111.8 |

Table 3: Total Reward on Synthetic Dataset. Oracle represents the best possible reward with perfect knowledge of the demand function.

## 5.2 MULTI-LABEL DATASETS

We use three multi-label datasets from LIBSVM repository (Elisseeff & Weston, 2002; Boutell et al., 2004; Chang & Lin, 2011), which are used for semantic scene, text and gene classification. We convert the supervised learning datasets to bandit feedback by creating a logging policy using 5% of the data following Swaminathan & Joachims (2015a); Lopez et al. (2020). More specifically, each feature $x$ has a label $y \in \{0, 1\}^p$ where $p$ is the number of labels. After the logging policy selects a label (action) $i$, a reward $y_i$ is revealed as bandit feedback $(x, i, y_i)$, *i.e.*, , for each data point, if the policy selects one of the correct labels of that data point, it gets a reward of 1 and 0 otherwise. By doing so, we have the full knowledge of counterfactual outcomes for evaluation. Data statistics are summarized in Section A.6 in Appendix.

Hamming loss, multi-label soft margin loss and reward are reported in Table 4, 5 and 6 respectively. CST-RI generally achieves comparable or superior performance against all baselines in all three datasets. Since we assume we do not know the logging policy, BanditNet performs poorly in dataset like Scene, which is consistent with Kang et al. (2007). HSIC has a comparable performance with CST-RI on TMC and Yeast, but performs poorly on Scene. We suspect this is due to the bias introduced in cross-validation which in turn results in a sub-optimal hyperparameter selection. UDM can improve over DM effectively but CST-RI still outperforms it significantly. Overall, CST-RI shows the most robust performance across all three metrics being studied.

## 5.3 RUNNING TIME ANALYSIS

We compare the average running time for one repetition for each experiment under same number of epochs. The results are summarized in Section A.4 in Appendix. Unsurprisingly, DM is the fastest algorithm. While our method is almost twice as slow as DM, it is still relatively fast compared to

|  | TMC | Yeast | Scene |
|---|---|---|---|
| Direct Method | 0.1224±0.0194 | 0.3669±0.0028 | 0.2428±0.0035 |
| CST-RI | **0.0775**±0.0023 | **0.2796**±0.0019 | **0.2306**±0.0372 |
| CST-AI | 0.5589±0.0336 | 0.3179±0.0580 | 1.2288±0.2255 |
| HSIC | 0.0797±0.0016 | 0.3024±0.0047 | 0.4504±0.0541 |
| UDM | 0.0905±0.0039 | 0.3547±0.0007 | 0.2293±0.0019 |

Table 4: Hamming Loss on Real Datasets

|  | TMC | Yeast | Scene |
|---|---|---|---|
| Direct Method | 2.0578±0.2034 | 9.2488±0.0137 | 5.1004±0.1019 |
| CST-RI | 0.2325±0.0222 | **0.5573**±0.0048 | **0.4918**±0.0512 |
| CST-AI | 3.0167±0.0682 | 0.9718±0.1691 | 1.2288±0.2255 |
| HSIC | **0.2259**±0.0037 | 0.5898±0.0019 | 0.9403±0.2616 |
| UDM | 1.3178±0.1877 | 7.0741±0.0243 | 3.8074±0.0444 |

Table 5: Multi-Label Soft-Margin Loss on Real Datasets

|  | TMC | Yeast | Scene |
|---|---|---|---|
| Direct Method | 3926.5±106.2 | 457.8±4.7 | 587.0±3.6 |
| CST-RI | **5035**±162.0 | **677.8**±1.6 | **604.0**±55.3 |
| CST-AI | 194.0±49.1 | 226.6±30.3 | 264.0±17.9 |
| HSIC | 4874.0±41.3 | 675.4±1.9 | 277.8±122.2 |
| BanditNet | 4302.3.4±84.2 | 574.8±8.6 | 243.4±52.0 |
| UDM | 4408.3±19.3 | 480.6±2.7 | 584.6±3.2 |

Table 6: Reward on Real Datasets

the other baselines. BanditNet is relatively slow due to the cross validation selection. Note that the time efficiency of HSIC is bottlenecked by its high computational complexity (Lopez et al., 2020). We thus observe HSIC is approximately 30 to 100 times slower than CST across all datasets. Since CST offers a competitive performance against HSIC with a much faster running time, it is potentially more suitable for large-scale applications which require frequent model update, such as a daily updated pricing system. For example, HSIC may take days for model re-training but CST can be updated day-to-day.

## 6 CONCLUSION AND FUTURE WORK

In this paper, we proposed a novel counterfactual self-training algorithm for learning from observational data. Comparing to existing approaches, our method is easy to compute and optimize. It also does not have the need for hyperparameter selection through cross validation, which is biased in nature for observational data. We provided a theoretical analysis showing self-training objective serves as an upper bound of the true risk of a randomized trial. However, our CST framework has several limitations. First, CST requires finite *discrete action set*. In order to augment observation data, CST will augment every action not observed. For continuous action, discretization or joint kernel embedding proposed in Zenati et al. (2020) might be used as an extension to CST, which we leave for future work. Second, CST in this paper can only work with *discrete outcomes*. If the outcome is continuous, it is also possible to extend our framework to continuous valued problems by: (1) discretize continuous value into discrete categories; (2) the pseudo-labels can be defined as self-ensemble (French et al., 2017) predictions, e.g. dropout (Bayesian neural networks) ensemble or temporal ensembling (Laine & Aila, 2016).

While this analysis is tailored for counterfactual learning, we hope it can shed light on a broader range of problems such as unsupervised domain adaptation and semi-supervised learning. It may also open doors for solving counterfactual learning with a model-based extrapolation for direct method. As shown in our pricing example, a good demand model may not necessarily lead to the highest revenue because of the downstream revenue maximization optimization (Elmachtoub & Grigas, 2017). A different formulation of target domain may help address this problem, which we leave for future work. Moreover, we believe our counterfactual self-training framework can be adapted to yield many specific algorithms for tasks such as learning from observational data with structured reward (Lopez et al., 2020; Kallus, 2019) and deficient historical logging policy (Sachdeva et al., 2020).

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

## A  APPENDIX

### A.1  PROOF OF THEOREM 1

**Theorem 1.** *Assume $f_\theta(r|x,p) \geq \frac{1}{M_0+1}$, where $M_0 > 1$ is constant, $x, p$ is defined on a compact, discrete set respectively, let $M = \min\{\max_{x,p}(\frac{\mathbb{P}}{f_\theta} - 1), M_0\}$, $f^\star = \operatorname{argmin}_{f \in \mathcal{F}} \mathcal{R}_\mathcal{D}(f)$, $\hat{\mathcal{D}}$ is the dataset generated by random imputation of current model output $f_\theta$, and $\hat{f}$ minimizes the empirical risk on $\hat{\mathcal{D}}$. For any loss function $l(\cdot, \cdot)$, we have:*

$$\mathcal{R}_\mathcal{D}(\hat{f}) - \mathcal{R}_\mathcal{D}(f^\star) \leq C(\sqrt{\frac{V}{n}} + \sqrt{\frac{\log(1/\delta)}{n}}) + (M+1)\hat{\mathcal{R}}_{\hat{\mathcal{D}}}(\hat{f}) - \mathcal{R}_\mathcal{D}(f^\star) \qquad (2)$$

*Proof.*

$$\mathcal{R}_\mathcal{D}(\hat{f}) - \mathcal{R}_\mathcal{D}(f^\star)$$

$$= \mathcal{R}_{\hat{\mathcal{D}}}(\hat{f}) - \mathcal{R}_{\hat{\mathcal{D}}}(f^\star) + \mathcal{R}_\mathcal{D}(\hat{f}) - \mathcal{R}_{\hat{\mathcal{D}}}(\hat{f}) - (\mathcal{R}_\mathcal{D}(f^\star) - \mathcal{R}_{\hat{\mathcal{D}}}(f^\star)) \qquad (4)$$

$$= \mathcal{R}_{\hat{\mathcal{D}}}(\hat{f}) - \mathcal{R}_{\hat{\mathcal{D}}}(f^\star) + \mathbb{E}_{\hat{\mathcal{D}}}(\frac{\mathbb{P}(r|x,p)}{f_\theta(r|x,p)} - 1)(l(\hat{f}(x,p),r) - l(f^\star(x,p),r)) \qquad (5)$$

$$\leq \mathcal{R}_{\hat{\mathcal{D}}}(\hat{f}) - \mathcal{R}_{\hat{\mathcal{D}}}(f^\star) + M\mathbb{E}_{\hat{\mathcal{D}}}l(\hat{f}(x,p),r) + \mathbb{E}_{\hat{\mathcal{D}}}l(f^\star(x,p),r) - \mathcal{R}_\mathcal{D}(f^\star) \qquad (6)$$

$$= (M+1)(\mathcal{R}_{\hat{\mathcal{D}}}(\hat{f}) - \hat{\mathcal{R}}_{\hat{\mathcal{D}}}(\hat{f})) + (M+1)\hat{\mathcal{R}}_{\hat{\mathcal{D}}}(\hat{f}) - \mathcal{R}_\mathcal{D}(f^\star) \qquad (7)$$

$$\leq C(\sqrt{\frac{V}{n}} + \sqrt{\frac{\log(1/\delta)}{n}}) + (M+1)\hat{\mathcal{R}}_{\hat{\mathcal{D}}}(\hat{f}) - \mathcal{R}_\mathcal{D}(f^\star) \qquad (8)$$

Equation 4 comes from adding and subtracting the risk defined on $\mathcal{D}$ and $\hat{\mathcal{D}}$. Since $\hat{\mathcal{D}}$ is imputed with probability $\mathbb{P}_\theta$, we can use importance sampling to get Equation 5. We get Equation 7 by adding and subtracting the empirical risk. Equation 8 is from the basic excess-risk bound. $V$ is the VC dimension of hypothesis class $\mathcal{F}$, and $C$ is a universal constant. $\qquad\square$

## A.2 PROOF OF PROPOSITION 1

**Proposition 1.** *CST with argmax imputation is convergent under certain conditions.*

*Proof.* Our CST objective is defined as

$$\min_{\theta,\hat{\mathbf{R}}} C_1 = \frac{1}{N|\mathcal{P}|}\Big(\sum_{i=1}^N l(f_\theta(x_i,p_i),r_i) + \sum_{i=1}^N \sum_{p\in\mathcal{P}\backslash p_i} l(f_\theta(x_i,p),\hat{r}_{i,p})\Big) \tag{9}$$

where $r_i$ is the factual data observed and $\hat{r}_{i,p}$ is imputed by trained classifier $f_\theta$. We show our proof using binary cross entropy loss which we use in the paper, it can be generalized to cross entropy easily. We show the convergence of CST-AI defined in Section 5, which imputes pseudo-label using argmax operation. The objective is optimized via the following two steps:

**1) Pseudo-Label Imputation:** Fix $\theta$ and impute $\hat{\mathbf{R}}$ to solve:

$$\min_{\hat{\mathbf{R}}} \sum_{i=1}^N \sum_{p\in\mathcal{P}\backslash p_i} -\Big(\hat{r}_i \log f_\theta(x_i,p) + (1-\hat{r}_i)\log(1-f_\theta(x_i,p))\Big) \tag{10}$$

$$s.t. \quad \hat{r}_{i,p} \in \Delta, \forall i,p$$

where $\Delta$ is the possible discrete values of the outcome.

**2) Model Retraining:** Fix $\hat{\mathbf{R}}$ and solve the following optimization using gradient descent, where $l(\cdot,\cdot)$ is binary cross entropy loss:

$$\min_\theta \sum_{i=1}^N l(f_\theta(x_i,p_i),r_i) + \sum_{i=1}^N \sum_{p\in\mathcal{P}\backslash p_i} l(f_\theta(x_i,p),\hat{r}_{i,p}) \tag{11}$$

For CST-AI, we have:

**Step 1) is non-increasing:** (10) has an optimal solution which is given by pseudo-labels imputed by argmax operation with feasible set being all possible outcomes. As a result, (10) is non-increasing.

**Step 2) is non-increasing:** If one use gradient descent to minimize the loss defined in Equation 11. The loss is guaranteed to decrease monotonically with a proper learning rate (Zou et al., 2019). For mini-batch gradient descent commonly used in practice, the loss is not guaranteed to decrease but also almost certainly converge to a lower value.

Since loss in Equation 9 is lower bounded, CST-AI is convergent.

$\qquad\square$

## A.3 EMPIRICAL CONVERGENCE ANALYSIS OF CST-RI

For CST-RI, since the convergence analysis is more challenging, we show empirically CST-RI converges in all of our experiments without any additional techniques. We show our loss curves in all experiments for our synthetic and multi-label datasets in Figure 7 and 11 respectively. CST-RI is trained with gradient descent with momentum (Ruder, 2016). For syntheic datasets, we set learning rate as 1e-3 and momentum as 0.9. For multi-label datasets, we set learning rate as 1e-1 and momentum as 0.9.

Next, we share some intuition on CST-RI and its connection with entropy regularization (Grandvalet & Bengio, 2005). Consider the second term in Equation 9 with stochastic imputation:

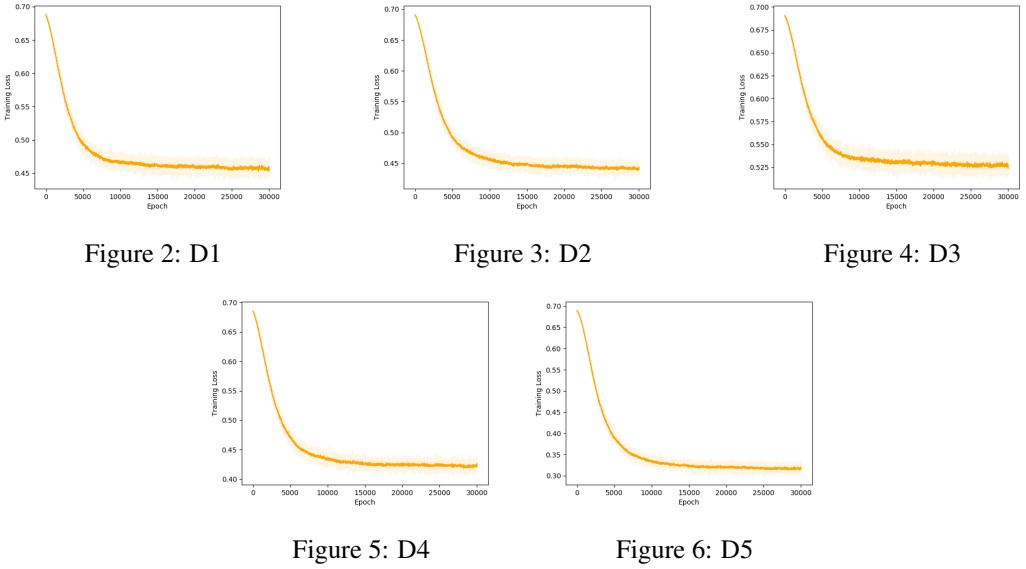

Figure 2: D1          Figure 3: D2          Figure 4: D3

Figure 5: D4          Figure 6: D5

Figure 7: Loss Curves for Synthetic Datasets

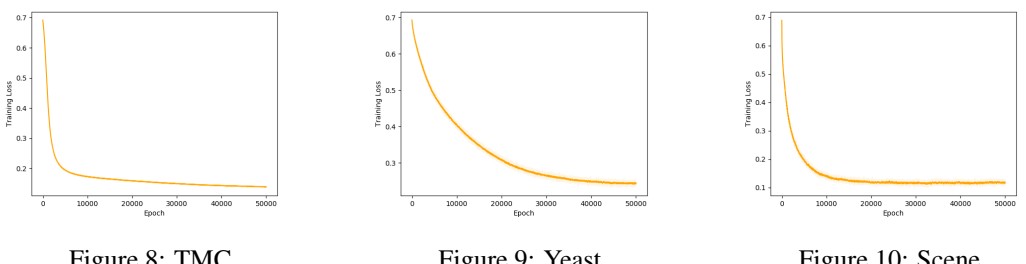

Figure 8: TMC          Figure 9: Yeast          Figure 10: Scene

Figure 11: Loss Curves for Multi-Label Datasets

$$\sum_{i=1}^{N} \sum_{p \in \mathcal{P} \backslash p_i} -\Big(\hat{r}_i \log f_\theta(x_i, p) + (1 - \hat{r}_i) \log(1 - f_\theta(x_i, p))\Big) \tag{12}$$

$$\hat{r}_{i,p} \sim Bern(f_\theta(x_i, p)), \forall i, p$$

Since $r \in \{0, 1\}$, by taking expectation over $\hat{r}$,

$$\mathbb{E}_{\hat{r}} \sum_{i=1}^{N} \sum_{p \in \mathcal{P} \backslash p_i} -\Big(\hat{r}_i \log f_\theta(x_i, p) + (1 - \hat{r}_i) \log(1 - f_\theta(x_i, p))\Big) \tag{13}$$

$$= \sum_{i=1}^{N} \sum_{p \in \mathcal{P} \backslash p_i} -\Big(f_\theta(x_i, p) \log f_\theta(x_i, p) + (1 - f_\theta(x_i, p)) \log(1 - f_\theta(x_i, p))\Big) \tag{14}$$

which equals to the entropy term defined on $f_\theta$, thus Our CSI-RI framework can be viewed as a variant of entropy regularization in semi-supervised learning (Grandvalet & Bengio, 2005). Since we aim to simulate a randomized trail, the hyper-parameter in Grandvalet & Bengio (2005) is set to 1 in CST. Instead of taking the argmax imputation which is commonly used in classification EM (CEM) (Amini & Gallinari, 2002) that minimizes the objective, we impute a randomly assigned

label with a larger probability to be the CEM solution. This step is very similar to deterministic annealing EM (Grandvalet & Bengio, 2005; Yuille et al., 1994; Rose et al., 1990) where a pseudo-label is generated by the output probability with a annealing temperature instead of CEM solution, which aims to find the global minimum more efficiently.

## A.4 EXPERIMENT RESULTS FOR RUNNING TIME ANALYSIS

|  | D1 | D2 | D3 | D4 | D5 |
|---|---|---|---|---|---|
| Direct Method | 21.64±1.46 | 21.34±1.81 | 21.70±1.71 | 21.44±1.66 | 21.67±1.18 |
| CST-RI | 38.28±1.23 | 35.30±0.02 | 36.65±0.27 | 37.19±0.33 | 37.71±0.80 |
| CST-AI | 38.00±1.14 | 35.90±0.04 | 36.99±0.26 | 38.10±0.54 | 37.89±0.33 |
| HSIC | 4038.35±4.65 | 4029.02±0.51 | 4087.17±1.36 | 4060.01±0.79 | 4055.30±1.11 |
| BanditNet | 657.82±10.50 | 614.91±1.99 | 632.93±2.77 | 639.83±1.53 | 659.10±3.11 |
| UDM | 24.99±0.34 | 25.85±0.91 | 25.18±0.39 | 24.23±0.49 | 25.57±0.48 |

Table 7: Running Time on Synthetic Datasets (measured in seconds)

|  | TMC | Yeast | Scene |
|---|---|---|---|
| Direct Method | 570.63±0.87 | 33.43±2.95 | 26.88±0.29 |
| CST-RI | 1428.65±2.22 | 93.60±4.14 | 71.98±0.69 |
| CST-AI | 1441.3±3.00 | 95.11±4.54 | 72.74±0.59 |
| HSIC | 43806.3±216.05 | 3276.82±46.25 | 3299.99±18.90 |
| BanditNet | 10342.65±35.83 | 488.71±35.84 | 271.22±8.79 |
| UDM | 958.49±0.99 | 39.03±0.11 | 32.20±0.04 |

Table 8: Running Time on Real Datasets (measured in seconds)

## A.5 DATA GENERATION FOR SYNTHETIC DATASET

In the synthetic experiments, we use a pricing example similar to the experiment in Lopez et al. (2020). Let $U(\cdot, \cdot)$ be a uniform distribution. Assume customer features are a 50-dimensional vector $X$ drawn from $U(0, 1)^{50}$ and there are 10 price options from \$1 to \$10. The logging policy is set as $\pi(p = i|x) = \frac{x_i}{\sum_{i=1}^{10} x_i}$. $\sigma$ denotes sigmoid function. We simulated five types of demand functions, with $h(x) = \sum a_i \sum \exp(\sum b_j \|x_j - c_j\|)$, $a, b, c \sim U(0, 1)^{50}$, $r \in \{0, 1\}$ :

- $r \sim \sigma(h(x) - 2x_0 \cdot p)$
- $r \sim \sigma(5 \cdot (x_0 - 0.5) - 0.4 \cdot p)$
- $r \sim \sigma(h(x) - \text{stepwise1}(x_0) \cdot p)$
- $r \sim \sigma(h(x) - \text{stepwise2}(x_0, x_1) \cdot p)$
- $r \sim \sigma(h(x) - (x_0 + x_1) \cdot p)$

where the stepwise function is defined as:

$$stepwise1(x) = \begin{cases} 0.7, & \text{if } x \leq 0.1 \\ 0.5, & \text{if } 0.1 < x \leq 0.3 \\ 0.3, & \text{if } 0.3 < x \leq 0.6 \\ 0.1, & \text{if } 0.6 < x \leq 1 \end{cases} \tag{15}$$

$$stepwise2(x, y) = \begin{cases} 0.65, & \text{if } x \leq 0.1 \text{ and } y > 0.5 \\ 0.45, & \text{if } x \leq 0.1 \text{ and } y \leq 0.5 \\ 0.55, & \text{if } 0.1 < x \leq 0.3 \text{ and } y > 0.5 \\ 0.35, & \text{if } 0.1 < x \leq 0.1 \text{ and } y \leq 0.5 \\ 0.45, & \text{if } 0.3 < x \leq 0.6 \text{ and } y > 0.5 \\ 0.25, & \text{if } 0.3 < x \leq 0.6 \text{ and } y \leq 0.5 \\ 0.35, & \text{if } 0.6 < x \leq 1 \text{ and } y > 0.5 \\ 0.15, & \text{if } 0.6 < x \leq 1 \text{ and } y \leq 0.5 \end{cases} \tag{16}$$

## A.6 MULTI-LABEL DATASETS STATISTICS

|       | # features | # labels | train size | test size |
|-------|------------|----------|------------|-----------|
| Yeast | 103        | 14       | 1500       | 917       |
| Scene | 294        | 6        | 1211       | 1196      |
| TMC   | 30438      | 22       | 21519      | 7077      |

Table 9: Dataset Statistics

