# OpenReview forum: "Counterfactual Self-Training"
_ICLR.cc/2021/Conference — Reject_

### Official Review · AnonReviewer1 · 2020-10-24
**Many details have not been discussed; I think the current version requires a major revision.**

**Rating:** 4
**Confidence:** 5

**Review:**

Post-Rebuttal:

I would like to thank the authors for their rebuttal.
The updated version of the paper has addressed some of my comments.
However, I still fail to understand why the method should 1) converge in general, and 2) converge to a good solution.
I have updated my score accordingly.

=====================================================================================================

Original Review:

Summary:
This paper proposes a method for causal inference given observational data. The authors formulate this task as a domain adaptation problem and propose a self-training algorithm to address it. The algorithm imputes the counterfactual outcomes (to simulate a randomized controlled trial) in an iterative process that supposedly gets better and better at estimating treatment outcomes.

I have many concerns about the paper; I touch on some of them in the following:
- There seems to be a misunderstanding in the literature review. There are two main approaches for *Policy Optimization*: Direct Method (DM) and Utility Maximization (UM). Counterfactual Risk Minimization (CRM) is a method in the UM category. It is the *Causal Inference* task that this paper addresses; not *Policy Optimization*.
- It is unclear why the proposed algorithm, namely Counterfactual Self-Training, should perform well; i.e., why a model trained on a new dataset with randomly imputed counterfactuals should do any better …
- It seems that the second term in the objective function in Eq. (1) is always zero; since, $\hat{r_{i, p}}$ is equal to $f_{\theta}(x_i, p)$ according to the line just above Eq. (1). Therefore, the objective function reduces to an unweighted factual loss, which we know would not result in a good model, since it does not account for selection bias.
- Algorithm 1, on the other hand, says that $\hat{r_{i, p}}$ is sampled from $P_{\theta}( r | x_i, p )$. The authors should comment on whether this is equivalent to $f_{\theta}(x_i, p)$...?
- The iterative process of imputing counterfactuals and then training a new model on the new RCT-like dataset is said to continue until the optimization converges. However, the authors do not discuss what the convergence criteria are.
- Definition for $M$ in Theorem 1 is vague: which values for $P$ and $P_{\theta}$ are used? Shouldn’t $M$ be a constant?


Minor comment(s):
- There are many typos in the paper, some of which are in the equations.

---

> ### Author Response · Authors · 2020-11-17
> **Responses to Reviewer 1:**
>
> -Policy Optimization and Causal Inference:
> Thanks for the suggestion! We have made it clearer in the related work and referred the two kinds of algorithms as counterfactual policy optimization. Also, as mentioned by Reviewer 4, direct method is one type of S-learner [9] that can be used both in estimating conditional average treatment effect and policy optimization, CST can be considered as an extension to direct method.
>
> -Why self-training works?
> First, in [7,8], self-training is formulated as a form of entropy regularization which can be considered as a measure of class overlap. This is related to the usefulness of unlabeled data where labeling is indeed ambiguous. We also added in some discussion on the connection between CST-RI with entropy regularization in Section A.3 in the updated submission. Second, we offer an intuition in the paper of transforming observational data into randomized trial and a theoretical motivation to show that the self-training objective is an upper bound of the risk on a randomized trial. Last but not least, to illustrate why unlabeled data is helpful, we provide an intuitive example used in [4] for a document classification task: ``Suppose, for example, that using only the labeled data we determine that documents containing the word 'homework' tend to belong to the positive class.  If we use this fact to estimate the classification of the many unlabeled documents, we might find that the word 'lecture' occurs frequently in the unlabeled examples that are now believed to belong to the positive class. This co-occurrence of the words 'homework' and 'lecture' over the large set of unlabeled training data can provide useful information to construct a more accurate classifier that considers both 'homework' and 'lecture' as indicators of positive examples.'' Another useful example is the X-learner in [9] mentioned by Reviewer 4.
>
> -Eq 1 second term equals 0:
> The second term in Eq (1) is not zero since $r$ is categorical, not continuous. We would like to emphasize that we consider $r$ as discrete labels, which we have made clearer in the updated submission. In fact, pseudo-labels are commonly used in self-training, and are usually defined as the argmax onehot of the model prediction on target data [1], in which case the loss would not be zero unless the model always makes 100\% confident predictions (which also means the model converges and self-training is done). In our case, $\hat{r}$ is sampled from a Bernoulli distribution $P_{\theta}(r|x_i,p)$ and it is easy to verify that the loss is not zero. Selection bias is already considered, please see our rebuttal to Reviewer 4. We would like to point out that it is also possible to extend our framework to continuous valued problems by: (1) discretize continuous value into discrete categories; (2) the pseudo-labels can be defined as self-ensemble [3] predictions, e.g. dropout (Bayesian neural networks) ensemble or temporal ensembling [2].
>
> -$P_\theta$ and $f_\theta$:
> $P_\theta(r|x_i, p)$ is equivalent to $f_\theta(r|x_i, p)$. We have made the notations more consistent across the text in the updated submission.
>
> -Convergence Result:
> Please see our response to Reviewer 4.
>
> -Definition of M:
> $M$ is defined as the maximum over $x$ and $p$, which is defined on compact and discrete sets respectively. We updated our submission to show it more clearly. $M$ is smaller than $M_0$, which is a constant.
>
> [1] Zou, Yang, et al. "Unsupervised domain adaptation for semantic segmentation via class-balanced self-training." Proceedings of the European conference on computer vision (ECCV). 2018.
>
> [2] Samuli Laine and Timo Aila. "Temporal ensembling for semi-supervised learning." In International Conference on Learning Representations (ICLR). 2017.
>
> [3] French, Geoff, Michal Mackiewicz, and Mark Fisher. "Self-ensembling for visual domain adaptation." International Conference on Learning Representations (ICLR). 2018.
>
> [4] Nigam, Kamal, et al. "Text classification from labeled and unlabeled documents using EM." Machine learning 39.2-3 (2000): 103-134.
>
> [5] Swaminathan, Adith, and Thorsten Joachims. "Counterfactual risk minimization: Learning from logged bandit feedback." International Conference on Machine Learning. 2015.
>
> [6] Lopez, Romain, et al. "Cost-Effective Incentive Allocation via Structured Counterfactual Inference." AAAI. 2020.
>
> [7] Grandvalet, Yves, and Yoshua Bengio. "Semi-supervised learning by entropy minimization." Advances in neural information processing systems. 2005.
>
> [8] Han, Ligong, et al. "Unsupervised domain adaptation via calibrating uncertainties." CVPR Workshops. Vol. 9. 2019.
>
> [9] Künzel, Sören R., et al. "Metalearners for estimating heterogeneous treatment effects using machine learning." Proceedings of the national academy of sciences 116.10 (2019): 4156-4165.
>
> [10] Zou, Yang, et al. "Confidence regularized self-training." Proceedings of the IEEE International Conference on Computer Vision. 2019.

---

### Official Review · AnonReviewer4 · 2020-10-28
**Counterfactual Self-Training**

**Rating:** 6
**Confidence:** 4

**Review:**

After Reading Authors' response.

Thanks the authors for the response. The additional discussion regarding convergence is appreciated. The explanation of how BanditNet is trained and the additional baseline makes the empirical comparison more clear. I would like to increase my evaluation to 6.

I still have the following concerns.

The generalization error bound is too loose to be informative.

There is no guarantee that the algorithm will continue to improve the performance.

------------------------------------------------------Original Comments----------------------------------------------------------

Summary

The paper proposed to use self-training, a semi-supervised learning technique to solve batch learning from biased and partial bandit feedback problem. The self-training process is as follows:

(1) Train a reward prediction model from observed data.
(2) Impute rewards for unobserved actions given a context with the reward prediction model.
(3) Train a reward prediction from the imputed rewards and go to step 2.

The paper provides generalization error analysis that connects performance of a policy under the uniform distribution to the objective they use for self-training.

The paper presents empirical analysis on both simulated data and multi-label classification dataset.

Strengths:

1. Using semi-supervised learning methods for batch learning from bandit feedback is interesting.

2. The paper is clearly written and easy to follow.


Weaknesses:

1. The proposed algorithm does not take the selection bias into account. This leads to a generalization error that depends on M, which can be very large in practice.

2. There is no guarantee that the algorithm will converge or continue to improve.

3. It would be great to compare with methods that model the selection bias, e.g. \sum\frac{1}{N}\frac{1}{\mu(p_i|x_i)} l( prediction_i ,r_i ) in Wang et al. Batch Learning from Bandit Feedback through Bias Corrected Reward Imputation. I understand that the paper assumes the logging policy is not available, but there are ways to estimate the logging policy, e.g. Hanna et.al. Importance Sampling Policy Evaluation with an Estimated Behavior Policy. Similarly, it seems not fair to use BanditNet but without modeling the selection bias.




Additional feedback

It would be great to mention that V is the VC dimension in the main paper.

In the generalization error bound, minimizing the right hand side does not equation 1 which uses the observed reward instead of imputed reward for actions selected by the logging policy.

---

> ### Author Response · Authors · 2020-11-17
> **Responses to Reviewer 4:**
>
>
> -The proposed method doesn't take selection bias into account:
> CST framework *does* take selection bias into account, albeit in a different fashion from IPS methods. In our data augmentation procedure, we only augment actions that were not selected by the historical logging policy. The end goal is to  simulate a randomized trial in the augmented dataset which consists of the factual data with the selection bias and the counterfactual data with imputed pseudo-labels.
>
> We only use M in the generalization bound to serve as our theoretical motivation, empirically we find CST has a robust performance across all datasets with a significant improvement over all baselines.
>
> -Convergence Result:
> Thanks for the feedback! We have added a new proposition in the updated submission for the convergence analysis of CST-AI and presented empirical evidence on the training loss curves in Section A.3 to demonstrate that CST-RI converges in all of our experiments without any additional technique using standard optimizers like gradient descent with momentum.
> Although CST-AI enjoys convergence in the added proposition, it shows inferior performance compared with CST-RI. We hypothesize this is because pseudo-labels can be noisy and self-training can put overconfident label belief on wrong categories. In fact, CST-AI usually comes with heuristic techniques such as self-paced training, class balancing [2] and other regularization tricks [10]. We consider the proposed random imputation (RI) as a useful regularization technique to prevent model getting too confident, and importantly, doesn't introduce additional hyper-parameters and cross-validation tuning. We indeed observe that RI regularization doesn't affect convergence in practice.
>
> -Comparison to BanditNet [3] and [2]:
> We estimated the logging policy (selection bias) $\rho(p_i|x_i)$ with an additional model for BanditNet as in [4], so we believe it is a fair comparison. We have updated the submission to describe the experimental setup more clearly.
>
> Thanks for the suggestion! We added the uniform direct method  in [2] as an additional baseline and included the result is in the updated submission. While this new baseline effectively improves over direct method, CST-RI still significantly outperforms it in all experiments.
>
> -VC dimension and Equation 1:
> We have added the description of V in the updated submission. There are two parts in Equation 1. The first part corresponds to observed reward and the second part corresponds to the imputed reward. The empirical risk of the combined dataset is $\hat{R}(\hat{D})$ which is on the right hand side.
>
> [1] Zou, Yang, et al. "Confidence regularized self-training." Proceedings of the IEEE International Conference on Computer Vision. 2019.
>
> [2] Wang, Lequn, et al. "Batch Learning from Bandit Feedback through Bias Corrected Reward Imputation."
>
> [3] Joachims, Thorsten, Adith Swaminathan, and Maarten de Rijke. "Deep learning with logged bandit feedback." International Conference on Learning Representations. 2018.
>
> [4] Lopez, Romain, et al. "Cost-Effective Incentive Allocation via Structured Counterfactual Inference." AAAI. 2020.

---

### Official Review · AnonReviewer2 · 2020-10-30

**Rating:** 5
**Confidence:** 3

**Review:**

The paper proposes to use self-training to tackle the fundamental problem of causal inference where only one potential outcome is seen. The proposed self-training method is iterative: after training a model on the observational dataset, they run points with different actions (treatments) through the trained model and collect the predictions, which are the pseudo-labels. They then continue the training of the model, including the pseudo-labels, until convergence. The paper experiments with two versions of the method -- one with deterministic pseudo-labels (CST-AI) and another with soft pseudo-labels sampled from a probability distribution (CST-RI). It is assumed that there are no unobserved confounders.

Although I liked the paper's proposal to bring self-training ideas to causal inference, the results don't seem fully convincing yet. I would have liked to see more baselines ( e.g. inverse propensity weighting), and perhaps experiments on more classic synthetic causal inference datasets (e.g. IHDP).

Results:
- Synthetic data: why the choice of hamming loss? RMSE and PEHE are more typical metrics here. The rather bad (and consistently bad) performance of CST-AI on the synthetic data is a little surprising. Do the authors have any explanation for why this is the case? Perhaps it was mentioned in the paper but I missed it.
- The run time analysis is interesting but its meaningfulness really depends on the setting in which the authors envision that this work could be used. Slow training time may be acceptable if test time inference is fast. As the authors pointed out, BanditNet is slow due to cross-validation, which other methods don't really do, so the comparison doesn't seem fair. Can the authors comment more on the practical significance of the run-time results?

References: in the classic causal inference literature, there are learners such as S, T, and X learners (this terminology is recent, but the models besides X learner have been around for a while; https://www.pnas.org/content/116/10/4156.abstract has some good explanations. The S learner learns a single model on observational data, then takes the data, switches the actions (treatment), and runs it through the model to collect the pseudo-labels. Training is not continued; treatment effects are simply estimated as a difference of the labels and pseudo-labels. But it also assume that pseudo-labels collected by training a model on observational data and switching actions are valid, so it might be interesting to cite some of this literature. Hill 2011 cited in the paper is actually a S-learner.

Minor points:
- I would suggest renaming "skyline" (e.g. in Table 3) which "represents the best possible reward with perfect knowledge of the demand function" (authors' words) to oracle, which is a more common term.

---

> ### Author Response · Authors · 2020-11-17
> **Response to Reviewer 2**
>
>
> -Why hamming loss:
> We apologize for the confusion. Our CST framework only works with finite discrete action set and finite discrete outcomes. Hence, we used hamming loss instead of RMSE. We have updated our paper to make our setup clearer and added discussions about the possible extensions of our framework to continuous settings.
> Since we are focused on multiple actions, PEHE is not suitable in our experiments.
> We discuss this limitation and possible extensions in our updated version of the submission to address this confusion. We are also happy to provide results for RMSE if Reviewer 2 feels it is necessary.
>
> -More results for IHDP:
> Since CST requires discrete outcome and IHDP [1] uses continuous outcome, IHDP is not suitable. We are happy to provide additional results for other suitable datasets.
>
> -Why CST-AI is worse than CST-RI:
> Good point. Argmax imputation (AI) is generally adopted in unsupervised domain adaptation problems. In practice, it usually comes with heuristic tricks such as self-paced training and class balancing [2] to prevent the model from converging too fast and getting too confident. Since cross-validation is biased in counterfactual setting due to selection bias in the validation set from historical logging policy [3], we try not to introduce additional hyper-parameters in our framework. We believe that the performance of CST-AI should improve if the aforementioned tricks were applied. We also consider random imputation as a useful regularization technique to prevent the model getting too confident.
>
> -Incorporate other baseline like inverse propensity weighting:
> BanditNet, which is one of the benchmarks used in our experiments, is one of the state-of-the-art methods using inverse propensity score matching. Since the logging policy is not known but estimated by an additional model, it does not show robust performance. Similar results have been shown in [4,5]. We also add an additional baseline in the updated submission using inverse propensity weighting to improve direct method [6], CST-RI still significantly outperforms this baseline.
>
> -Run time analysis:
> The run time analysis is used to highlight the computational advantage of CST, compared to SOTA baseline like HSIC. Faster running time can be helpful for deployment of updated models in a large-scale business applications. For example, HSIC may take days for re-training but CST can be updated on a daily basis. We thank the reviewer for this feedback, and we have added a discussion in the paper.
>
> -Reference and Skyline:
> Thanks for comment! We have added the reference in related work and changed the name of Skyline to Oracle.
>
> [1] https://git.lamsade.dauphine.fr/cbeji/causality/-/blob/2ecaa085b141ffb637dec4a092b4f3600a64f7cf/data/IHDP/
>
> [2] Zou, Yang, et al. "Unsupervised domain adaptation for semantic segmentation via class-balanced self-training." Proceedings of the European conference on computer vision (ECCV). 2018.
>
> [3] Lopez, Romain, et al. "Cost-Effective Incentive Allocation via Structured Counterfactual Inference." AAAI. 2020.
>
> [4] Noveen Sachdeva, Yi Su, and Thorsten Joachims. Off-policy bandits with deficient support. In Proceedings of the 26th ACM SIGKDD International Conference on Knowledge Discovery & Data Mining, pp. 965–975, 2020.
>
> [5] Joseph DY Kang, Joseph L Schafer, et al. Demystifying double robustness: A comparison of alternative strategies for estimating a population mean from incomplete data. Statistical science, 22(4):523–539, 2007.
>
> [6] Wang, Lequn, et al. "Batch Learning from Bandit Feedback through Bias Corrected Reward Imputation."

---

### Decision · Program_Chairs · 2021-01-07
**Final Decision**

**Decision:**

Reject

**Comment:**

The paper proposes an intriguing approach for "individual treatment effect" estimation from an observational dataset.  The approach is developed for multiple discrete actions (beyond binary treatments as typically studied in ITE literature) and discrete outcomes (a special case compared to related literature). The idea is to use the "direct method" (i.e. learn a probabilistic classifier using the observed dataset) and sample imputed outcomes for all unobserved action-outcomes. Then, learn a probabilistic classifier that fits the observed+imputed dataset well, and iterate the procedure. This intriguing idea seems to converge empirically on a few different problems, and sampling the imputations rather than using deterministic imputations seems to be an important detail.
Proof of convergence is however shown for deterministic imputations. The generalization error bound (Theorem 1) also does not show adequate motivation for the proposed method -- even with infinite data (n->infinity), the excess risk could scale with the empirical risk of the returned model on the imputed dataset. Without an additional step proving that empirical risk on hat{D} (the imputed dataset) converges to 0 during successive iterations of the procedure, the generalization error bound is incomplete.
Consider the example of Figure 1, but where customer A has arrived to the system twice. So, the dataset contains {x1, $2, 1} and {x1, $3, 0}. When constructing the imputed dataset, the first data-point would create 2 regression examples {x1, $1, ..} and {x1, $3, ..} while the second data-point would create 2 regression examples {x1, $1, ..} and {x1, $2, ..}.
Now, if the two {x1, $1, ..} examples have different imputation labels sampled from the model, this sets up an unrealizable learning problem and the empirical risk on hat{D} cannot be 0 for any predictor.
In this toy example, we might know that we should "collapse" the two data-points (e.g., de-duplicate the dataset to only have unique x's with aggregated action-outcomes across all observations) in the original data-set and only create one set of imputed labels -- but similar unrealizability can happen for x's that are "close" to each other that no model has capacity to label them differently.

The strength of the paper is its intriguing approach to ITE estimation. It is a form of an iterative S-learner (vanilla S-learners have been widely used in ITE estimation).
The low-point of the paper is this weakness in theory and analysis -- it is unclear if the proposed procedure with sampling imputations (which seems to be important for empirical performance) is even a consistent algorithm.
The paper would be much stronger with a more rigorous analysis of when the method will reliably work, and importantly, its limitations -- such a study will help practitioners know when to use self-training over direct method, targeted max likelihood, S-learners, etc.